# Global Trends and Future Directions in Agricultural Remote Sensing for Wheat Scab Detection: Insights from a Bibliometric Analysis

Sarfraz Hussain [1,†], Ghulam Mustafa [1,2,3,†] , Imran Haider Khan [2,3], Jiayuan Liu [1], Cheng Chen [1], Bingtao Hu [1], Min Chen [1], Iftikhar Ali [3] and Yuhong Liu [1,*]

[1] Key Laboratory of Integrated Regulation and Resource Development on Shallow Lakes, Ministry of Education, College of Environment, Hohai University, Nanjing 210098, China; 20210936@hhu.edu.cn (S.H.); 20220939@hhu.edu.cn or gmawan197@outlook.com (G.M.); ljy17695687200@hhu.edu.cn (J.L.); chencheng93@hhu.edu.cn (C.C.); 200205020006@hhu.edu.cn (B.H.); minchen@hhu.edu.cn (M.C.)

[2] National Engineering and Technology Center for Information Agriculture, Key Laboratory for Crop System Analysis and Decision Making, Ministry of Agriculture, Jiangsu Key Laboratory for Information Agriculture, Jiangsu Collaborative Innovation Center for Modern Crop Production, Nanjing Agricultural University, Nanjing 210095, China; 2018201104@njaue.edu.cn or imranhaiderkhan110@hotmail.com

[3] Department of Agronomy, University of Agriculture Faisalabad, Faisalabad 38000, Pakistan; dr.iftikharali@uaf.edu.pk

[*] Correspondence: yhliu@hhu.edu.cn

[†] These authors contributed equally to this work.

**Abstract:** The study provides a comprehensive bibliometric analysis of imaging and non-imaging spectroscopy for wheat scab (INISWS) using CiteSpace. Therefore, we underpinned the developments of global INISWS detection at kernel, spike, and canopy scales, considering sensors, sensitive wavelengths, and algorithmic approaches. The study retrieved original articles from the Web of Science core collection (WOSCC) using a combination of advanced keyword searches related to INISWS. Afterward, visualization networks of author co-authorship, institution co-authorship, and country co-authorship were created to categorize the productive authors, countries, and institutions. Furthermore, the most significant authors and the core journals were identified by visualizing the journal co-citation, top research articles, document co-citation, and author co-citation networks. The investigation examined the major contributions of INISWS research at the micro, meso, and macro levels and highlighted the degree of collaboration between them and INISWS knowledge sources. Furthermore, it identifies the main research areas of INISWS and the current state of knowledge and provides future research directions. Moreover, an examination of grants and cooperating countries shows that the policy support from the People's Republic of China, the United States of America, Germany, and Italy significantly benefits the progress of INISWS research. The co-occurrence analysis of keywords was carried out to highlight the new research frontiers and current hotspots. Lastly, the findings of kernel, spike, and canopy scales are presented regarding the best algorithmic, sensitive feature, and instrument techniques.

**Keywords:** wheat scab; agricultural remote sensing; knowledge map; CiteSpace; co-authorship; institution co-authorship

## 1. Introduction

Investigating a research field's comprehensive academic background and knowledge structure is a prolific method to determine the hotspots, research themes, knowledge foundations, and research frontiers in that particular research domain at a global scale. The comprehensive background of scientific studies can be classified by a series of different specialties, such as collaborating authors, institutions, countries, journals, co-occurring keywords, hot research topics, knowledge clusters, and cited references. Additionally, the

clustering of the literature can determine the knowledge structure and domains based on co-occurring keywords and research themes from the research papers downloaded from the databases [1,2]. These analyses, also referred to as bibliometric or scientometric analyses, can be conducted, and the networks can be visualized to extract knowledge maps [3,4]. This new method for analyzing scientific literature includes thorough and comprehensive interpretations of the intellectual background of almost any research field. Additionally, it aids in understanding cutting-edge research frontiers, author or institutional collaborations, knowledge structures, and novel developments that are important to engineers, business investors, and researchers. In this article, we emphasized the scab disease of the wheat crop in the context of agricultural remote sensing (ARS).

For the first time, scab was identified as a new wheat, barley, and ryegrass disease in England caused by *Fusisporium culmorum*, *hordei*, and *Lolii W. Sm*, respectively [5]. At the end of the 19th century, it was noted as a significant disease in the United States of America (USA). At the start of the 20th century, scab was well-known worldwide in wheat-producing regions [6]. McMullen et al. (2012) characterized it as a re-emerging disease due to the frequent epidemics of wheat in the USA and Canada from 1991 to 1996 [7]. China has experienced 30 fusarium head blight (FHB) epidemics since 1950, each affecting more than 10% of the country's land area. In 2012, a massive outbreak of wheat disease affected about 10 mha land of wheat production and resulted in a yield loss of more than 2 million tons [8]. Lower Yangtze River Valley and Heilongjiang Province are two of China's most frequently hit regions in the east. Also noteworthy is that damage has spread north and west, devouring the Huang-Huai River Valley, China's largest wheat-growing region. Approximately 17 percent of world wheat is produced in China, where scab has the most significant impact, causing yield losses of 10 to 20% in moderate epidemics and up to 50% in severe epidemics [9]. In addition, Japan, Korea, and the far east of Russia adjacent to Heilongjiang Province, China, were also severely affected by scab. The outbreak of 1963 in Japan affected 71.5% of the wheat acreage and resulted in a yield loss of 53.5% [10].

The involvement of two species from the genus *Microdochium* and multiple species from the genus *Fusarium* is referred to as a scab disease complex. The primary distinction between these two is that *Microdochium* species do not produce mycotoxins, while *Fusarium* species do [11]. In contrast, *Fusarium graminearum* is the most common pathogen of scab worldwide [7,12,13]. However, various studies revealed that other *Fusarium* species may significantly contribute to this disease in various parts of the world with various climatic conditions. For example, *F. graminearum*, *F. avenaceum*, *F. culmorum*, *F. tricinctum*, *F. poae*, and *M. majus* were Europe's dominant species [14,15]. While in Canada, *F. avenaceum*, *F. graminearum*, *F. poae*, *F. equiseti*, and *F. sporotrichioides* were the most frequent species during the last two decades [16]. FHB is a monocyclic disease; ascospores, macroconidia, and microconidia are all forms of the pathogen that can survive in the debris (Figure 1) of a previous crop within sexual structures called perithecia. These spores are regarded as the disease's primary inoculum. These spores are considered the main inoculum of the disease. Besides serving as hosts, both gramineous and non-gramineous weeds are known sources of the inoculum for scab. When weather conditions are favorable and in the anthesis stage, the inoculum is spread by wind or splashed by rain and lands on the open wheat kernels. The spores germinate on the spikelet tissue and form germ tubes [13,17].

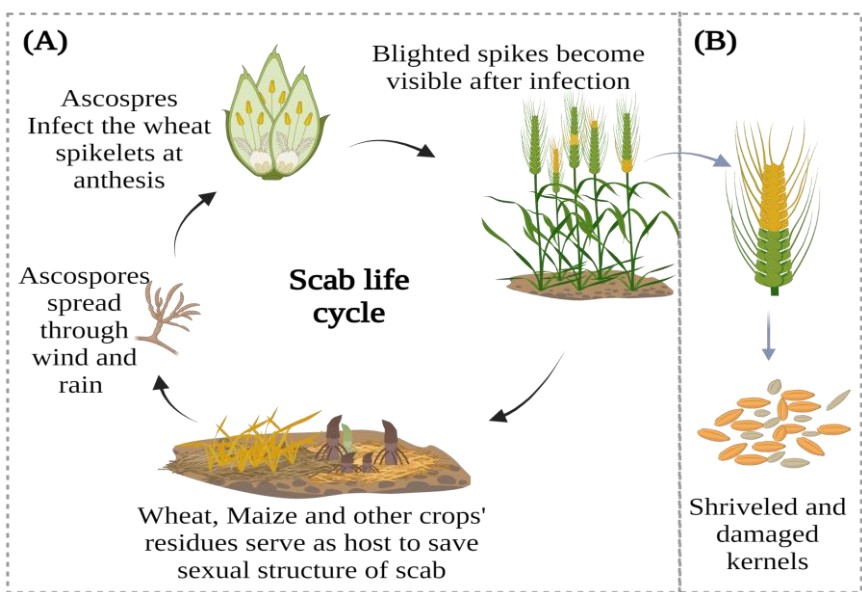

**Figure 1.** Illustration of the life cycle of scab (**A**) and damaged or shriveled grains (**B**), modified from [18].

Scab is critical as it causes colossal damage to crops—often causing damage or crop failure [19,20]. However, inappropriate fungicide application, control, and disease quantification employing inadequate measures are the leading factors causing substantial yield losses annually [21]. To this end, different optical sensor technologies have been introduced for several precision agriculture measures in phenotyping, phytopathology, and pedology [22–24]. The frontiers and hotspots of the study can explain the current scab study development under ARS advancements. This study conducted a bibliometric investigation of the advanced research conducted in ARS for wheat scab detection. The relevant scientific data published between 2000 and 2022 in high-quality journals were retrieved from the Web of Science core collection (WOSCC). The co-citation, co-occurrence, co-authorship, and cluster analyses were performed in CiteSpace. The objectives of the current study are as follows:

(a) Highlight the major research efforts in the domain of ARS for wheat scab detection at the level of contributing authors, institutions, and countries.
(b) Evaluate the contribution of key journals in the same area.
(c) Classify and interpret the obtained literature and knowledge into brief knowledge clusters using co-occurring keywords.
(d) Determine the research frontiers, knowledge foundation, and hot topics in the field of ARS for wheat scab detection for future studies.
(e) Review the conducted studies for scab detection at different scales (grains, spike, and canopy) of wheat crops.

## 2. Data Collection and Bibliometric Methodology

### 2.1. Retrieval of Data from Web of Science

The most precise literature-indexing resource is the Web of Science (WOS), which covers scientific and social, health, and economic knowledge. Therefore, worldwide WOS is frequently acknowledged as the best source of data collection for bibliometric analyses [2]. The WOS core collection (WOSCC) databases retrieved the relevant data. A large number of iterations were used to obtain an optimal searching keyword code to download the most relevant articles related to ARS for wheat scab detection. Table 1 shows a development series of iterative-searching keyword codes for probing WOS database data. The most effective searching keywords were as follows: ("wheat spike") (Topic) or ("fusarium head blight" OR "Scab") (Topic) and ("remote sensing") OR ("hyperspectral

imaging" OR "fluorescence" OR "reflectance" OR "hyperspectral reflectance") (Topic) and ("detection") OR ("classification") (Topic). It implies that the published documents were searched for contained words in the titles, abstracts, or keywords. Only peer-reviewed, original research articles published in English were extracted; review articles, books, and conference proceedings were omitted. The time frame of the data collection was from 1 January 2000 to 31 December 2022, inclusive. While downloading the pertinent literature, the research domains were limited to the sciences and technologies.

**Table 1.** Optimization of search keywords for finding WOS publications relevant to agricultural remote sensing for wheat scab detection.

| No. | Searching Code | Results | Quality |
|---|---|---|---|
| 1 | ("Wheat spike") (Topic) or ("Fusarium head blight") | 3560 | Very rough, very generic, highly irrelevant |
| 2 | ("Wheat spike") (Topic) or ("Fusarium head blight" OR "Scab") | 6825 | Improved, yet irrelevant |
| 3 | ("Wheat spike") (Topic) or ("Fusarium head blight" OR "Scab") (Topic) and ("remote sensing") | 163 | Very generic and highly irrelevant |
| 4 | ("Wheat spike") (Topic) or ("Fusarium head blight" OR "Scab") (Topic) and ("remote sensing") OR ("hyperspectral imaging") | 198 | Improved, yet irrelevant |
| 5 | ("Wheat spike") (Topic) or ("Fusarium head blight" OR "Scab") (Topic) and ("remote sensing") OR ("hyperspectral imaging" OR "Fluorescence") | 307 | A little improved, yet irrelevant |
| 6 | ("Wheat spike") (Topic) or ("Fusarium head blight" OR "Scab") (Topic) and ("remote sensing") OR ("hyperspectral imaging" OR "Fluorescence" OR "hyperspectral reflectance") | 338 | More improved, yet irrelevant |
| 7 | ("Wheat spike") (Topic) or ("Fusarium head blight" OR "Scab") (Topic) and ("remote sensing") OR ("hyperspectral imaging" OR "Fluorescence" OR "reflectance" OR "hyperspectral reflectance") (Topic) and ("detection") OR ("classification") OR ("monitoring") OR ("identification") (Topic) | 238 | Much improved, highly relevant. |

### 2.2. Schematic of the Study

Based on the methodology given above, a total of 238 original research articles were retrieved. The whole record and cited references were saved as "other file formats" results, and plain text was chosen as the file format. The schematic reveals the steps taken to continue the study in Figure 2.

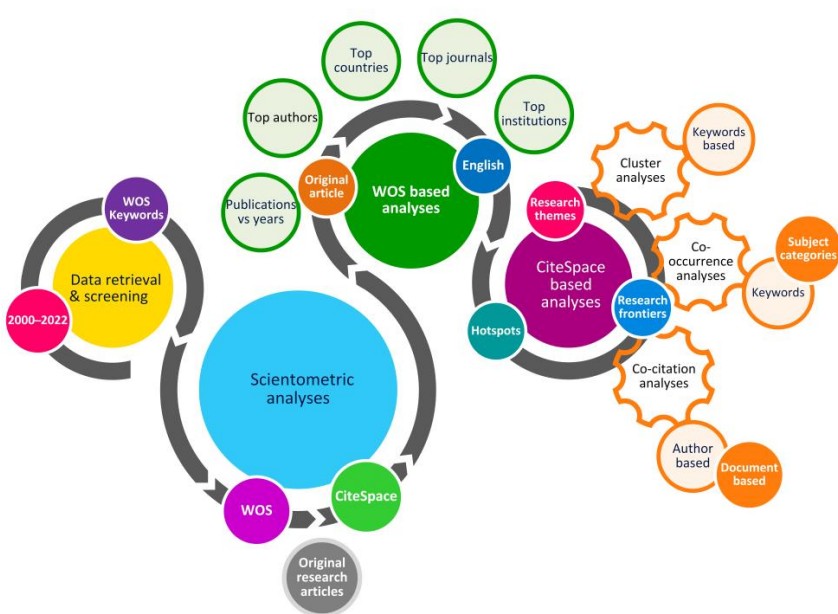

**Figure 2.** Step-by-step layout and schematic of the current study.

### 2.3. CiteSpace-Based Bibliometric Analyses

The more advanced capabilities for bibliometric analyses include a map or network analyses and the visualization of the scientific literature. CiteSpace is a Java-based program that maps and visualizes scientific domains for bibliometric analysis. Dr. Chen Chaomei developed this at Drexel University of United States [25], and it is freely available online. CiteSpace's advanced features and wide range of applications allowed for the visualization analyses of the literature maps in the field of ARS to detect wheat scab. The following steps were used to visualize the obtained data in CiteSpace. First, a project was created with the input of publications that were downloaded, complete with full records and cited references in plain text. Then, as shown in Table 2, subsequent parameters were established.

**Table 2.** CiteSpace parameters and values for bibliometric analysis of advanced research in agricultural remote sensing for wheat scab detection.

| No. | Parameters | Definition |
|-----|-----------|-----------|
| 1 | Time slicing | Year span from 2005 to 2022; years per slice of 1 year for all |
| 2 | Term source | Title, author, abstract, keywords, and keywords plus |
| 3 | Node type | Author, cited author, cited reference, institution, country, cited journal, and keywords |
| 4 | Selection criteria | Top 15% |
| 5 | Pruning | Pathfinder and pruning sliced networks |
| 6 | Links | Default |
| 7 | Visualization | Show merged network and cluster view-static |

After defining the parameters, the co-citation analysis and keywords co-occurrence analysis were run in CiteSpace to obtain the networks showing the co-citations among the authors, documents, and journals (co-citation analysis) and keywords, hot research topics, and research frontiers (keywords co-occurrence analysis), respectively. Finally, the relevant data and mapping networks were investigated, and the corresponding results of the current research study's visualization investigation were presented and discussed.

## 3. Examination and Interpretation of Scientometrics Analysis

### 3.1. Bibliometric Analyses Based on Web of Science

The distribution of corresponding citations and publications in the research area of ARS for wheat scab detection is plotted for a period from 1 January 2005 to 1 February 2022, as shown in Figure 3. It can be observed that a considerably slow rise was observed in the number of significant publications over the initial decade (years ranging from 2005 to 2010). A total of 13 studies were published in 2010, with a fall in the subsequent couple of years. From 2014, the concerns were significantly increased for the research. Compared to the statistics in 2020, the number increased in 2021, when a maximum of 34 studies were published. Relatively few numbers of publications in the area show that the ARS for wheat scab detection is still in its infancy, and significant efforts are required in the future to achieve further advancements in the field. The trends are almost similar for the citations, with a maximum of 724 in 2021.

The top 15 journals in which most articles related to ARS for wheat scab detection are published are enlisted against the number of publications and their percentage contribution, as shown in Table 3. Table 4 shows the top 15 institutions in the research domain of ARS for wheat scab detection. Tables 5 and 6 show the 15 most important countries and the most prolific authors, respectively, with the highest number of studies in the given research domain. The distribution of the top 7 funding agencies involved in the relevant articles extracted from the WOS in ARS research is exhibited in Table 7. Similarly, the top 15 WOS subject categories in the research domain of ARS for wheat scab detection are shown in Table 8.

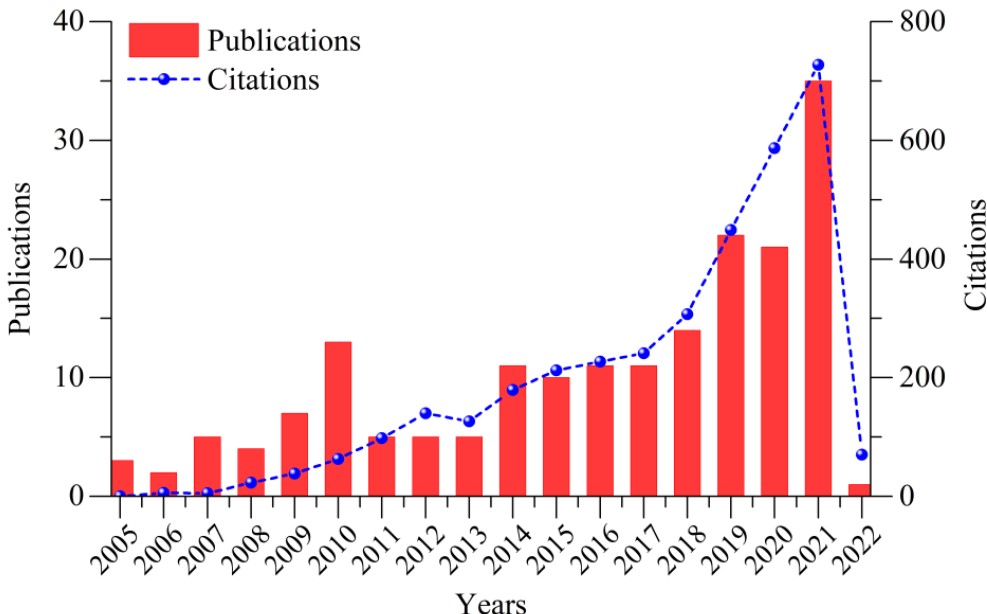

**Figure 3.** Publications and citations of total articles in agricultural remote sensing for wheat scab detection from 2005 to 2022.

**Table 3.** Top 15 journals in the research domain of agricultural remote sensing for wheat scab detection.

| No. | Journals | Records | % of Total |
|---|---|---|---|
| 1 | REMOTE SENSING | 14 | 7.568 |
| 2 | COMPUTERS AND ELECTRONICS IN AGRICULTURE | 7 | 3.784 |
| 3 | FRONTIERS IN PLANT SCIENCE | 7 | 3.784 |
| 4 | BIOSYSTEMS ENGINEERING | 6 | 3.243 |
| 5 | MOLECULAR BIOLOGY REPORTS | 4 | 2.162 |
| 6 | PHYTOPATHOLOGY | 4 | 2.162 |
| 7 | PLANT PATHOLOGY | 4 | 2.162 |
| 8 | SENSORS | 4 | 2.162 |
| 9 | BMC GENOMICS | 3 | 1.622 |
| 10 | CROP PASTURE SCIENCE | 3 | 1.622 |
| 11 | EUROPEAN JOURNAL OF PLANT PATHOLOGY | 3 | 1.622 |
| 12 | PLANT BIOTECHNOLOGY JOURNAL | 3 | 1.622 |
| 13 | PLANT DISEASE | 3 | 1.622 |
| 14 | PLANT PHYSIOLOGY | 3 | 1.622 |
| 15 | PLANTA | 3 | 1.622 |

**Table 4.** Top 15 institutions in the research domain of agricultural remote sensing for wheat scab detection.

| No. | Affiliations | Records | % of Total |
|---|---|---|---|
| 1 | CHINESE ACADEMY OF SCIENCES | 22 | 11.892 |
| 2 | ANHUI UNIVERSITY | 13 | 7.027 |
| 3 | UNITED STATES DEPARTMENT OF AGRICULTURE USDA | 12 | 6.486 |
| 4 | UNIVERSITY OF CHINESE ACADEMY OF SCIENCES CAS | 11 | 5.946 |
| 5 | CHINESE ACADEMY OF AGRICULTURAL SCIENCES | 8 | 4.324 |
| 6 | NORTHWEST A F UNIVERSITY CHINA | 8 | 4.324 |
| 7 | CHINA AGRICULTURAL UNIVERSITY | 7 | 3.784 |
| 8 | INSTITUTE OF CROP SCIENCES CAAS | 7 | 3.784 |
| 9 | INSTITUTE OF GENETICS DEVELOPMENTAL BIOLOGY CAS | 7 | 3.784 |
| 10 | INRAE | 6 | 3.243 |
| 11 | NANJING AGRICULTURAL UNIVERSITY | 6 | 3.243 |
| 12 | AGRICULTURE AGRI FOOD CANADA | 5 | 2.703 |
| 13 | BIOTECHNOLOGY AND BIOLOGICAL SCIENCES RESEARCH COUNCIL BBSRC | 5 | 2.703 |
| 14 | CONSEJO NACIONAL DE INVESTIGACIONES CIENTICAS Y TECNICAS CONICET | 5 | 2.703 |
| 15 | TUSCIA UNIVERSITY | 5 | 2.703 |

**Table 5.** Top 15 countries in the research domain of agricultural remote sensing for wheat scab detection.

| No. | Countries | Records | % of Total |
|---|---|---|---|
| 1 | CHINA | 73 | 39.459 |
| 2 | USA | 37 | 20.0 |
| 3 | GERMANY | 18 | 9.730 |
| 4 | ITALY | 12 | 6.486 |
| 5 | CANADA | 11 | 5.946 |
| 6 | FRANCE | 9 | 4.865 |
| 7 | ENGLAND | 7 | 3.784 |
| 8 | SOUTH KOREA | 7 | 3.784 |
| 9 | BELGIUM | 6 | 3.243 |
| 10 | ARGENTINA | 5 | 2.703 |
| 11 | AUSTRALIA | 5 | 2.703 |
| 12 | JAPAN | 4 | 2.162 |
| 13 | RUSSIA | 4 | 2.162 |
| 14 | BRAZIL | 3 | 1.622 |
| 15 | CZECH REPUBLIC | 3 | 1.622 |

**Table 6.** Top 15 authors in the research domain of agricultural remote sensing for wheat scab detection.

| No. | Authors | Records | % of Total |
|---|---|---|---|
| 1 | Huang WJ | 10 | 5.405 |
| 2 | Ma HQ | 9 | 4.865 |
| 3 | Dong YY | 8 | 4.324 |
| 4 | Liu LY | 7 | 3.784 |
| 5 | Huang LS | 6 | 3.243 |
| 6 | Cruz CD | 5 | 2.703 |
| 7 | Chen G | 4 | 2.162 |
| 8 | Chibbar RN | 4 | 2.162 |
| 9 | Favaron F | 4 | 2.162 |
| 10 | Gu CY | 4 | 2.162 |
| 11 | Hong MJ | 4 | 2.162 |
| 12 | Li LH | 4 | 2.162 |
| 13 | Schafer W | 4 | 2.162 |
| 14 | Sella L | 4 | 2.162 |
| 15 | Seo YW | 4 | 2.162 |

**Table 7.** Top 15 funding agencies in the research domain of agricultural remote sensing for wheat scab detection.

| No. | Funding Agencies | Records | % of Total |
|---|---|---|---|
| 1 | National Natural Science Foundation of China NSFC | 53 | 22.269 |
| 2 | National Key Research and Development Program of China | 21 | 8.824 |
| 3 | National Key R D Program of China | 9 | 3.782 |
| 4 | UK Research Innovation | 9 | 3.782 |
| 5 | Biotechnology and Biological Sciences Research Council | 7 | 2.941 |
| 6 | China Postdoctoral Science Foundation | 7 | 2.941 |
| 7 | National Basic Research Program of China | 7 | 2.941 |
| 8 | Youth Innovation Promotion Association Cas | 6 | 2.521 |
| 9 | Beijing Nova Program of Science and Technology | 5 | 2.101 |
| 10 | Chinese Academy of Sciences | 5 | 2.101 |
| 11 | Natural Sciences and Engineering Research Council of Canada | 5 | 2.101 |
| 12 | United States Department of Agriculture | 5 | 2.101 |
| 13 | Canada Research Chairs | 4 | 1.681 |
| 14 | Deutscher Akademischer Austausch Dienst Daad | 4 | 1.681 |
| 15 | French National Research Agency | 4 | 1.681 |

**Table 8.** Top 15 WOS subject categories in the research domain of agricultural remote sensing for wheat scab detection.

| No. | WOS Subject Categories | Records | % of Total |
|---|---|---|---|
| 1 | Plant Sciences | 64 | 34.595 |
| 2 | Agriculture Multidisciplinary | 29 | 15.676 |
| 3 | Agronomy | 29 | 15.676 |
| 4 | Food Science Technology | 19 | 10.27 |
| 5 | Geosciences Multidisciplinary | 15 | 8.108 |
| 6 | Remote Sensing | 15 | 8.108 |
| 7 | Environmental Sciences | 14 | 7.568 |
| 8 | Imaging Science Photographic Technology | 14 | 7.568 |
| 9 | Biochemistry Molecular Biology | 12 | 6.486 |
| 10 | Biotechnology Applied Microbiology | 12 | 6.486 |
| 11 | Genetics Heredity | 12 | 6.486 |
| 12 | Horticulture | 10 | 5.405 |
| 13 | Agricultural Engineering | 9 | 4.865 |
| 14 | Chemistry Applied | 9 | 4.865 |
| 15 | Computer Science Interdisciplinary Applications | 7 | 3.784 |

*3.2. Co-Citation Analysis*

A co-citation relationship exists among two or more authors or documents if they are cited simultaneously by a third author or document [26]. CiteSpace performed three basic types of co-citation analyses to identify documents' relationship and mapping structures, co-citing authors and journals. The co-citation analysis is a powerful tool for determining the degree of inter-relationship between journals, authors, and articles by creating a mapping structure and tracking the progress of scientific research fields [27].

3.2.1. Document Co-Citation Analysis

The articles or documents are the major constituents of the repository or databases of knowledge in ARS for wheat scab detection. The reference co-citation analysis or document is an effective method to assess the mapping and evolution of any research area [28]. A visualization network for cited documents was developed after the scientometric analysis in CiteSpace was run, as shown in Figure 4. The connections between the nodes serve as indicators of the co-citation relationships between the references or cited articles, whereas the nodes represent the cited documents. A larger node represents a more important document, and documents frequently cited by other documents are closely linked. Figure 4

shows that Alisaac et al. (2018) and Jin et al. (2018) are the most important studies conducted in the domain. The modularity Q and weighted mean silhouette S were 0.6262 and 0.8854, respectively, for the document co-citation analysis.

**Figure 4.** The visualization map for the document co-citation network of literature related to agricultural remote sensing for wheat scab detection.

The topmost fifteen cited articles are presented in Table 9, along with order and position (considering citation counts), citation counts, first author, publication year, journal name, volume number, pages, and DOIs. It can be observed that Jin et al. (2018) and Alisaac et al. (2018) were the leading authors with the highest co-citations, 12 and 11, respectively. Some other distinguished authors (and their co-citations) were Barbedo (10), Ropelewska (8), Zhang (8), etc. The knowledge structure of ARS for wheat scab detection can be gradually reshaped with the help of document co-citation analysis, as it facilitates the determination of the highly cited documents and significant research articles, which comprise the knowledge databases or domains of the field under consideration.

**Table 9.** Top 15 effective co-cited documents on agricultural remote sensing for wheat scab detection.

| Sr. No. | Count | Year | Cited References |
|---------|-------|------|------------------|
| 1 | 12 | 2018 | [29] |
| 2 | 11 | 2018 | [23] |
| 3 | 10 | 2015 | [30] |
| 4 | 8 | 2018 | [31] |
| 5 | 8 | 2019 | [32] |
| 6 | 7 | 2020 | [9] |
| 7 | 7 | 2018 | [33] |
| 8 | 7 | 2019 | [34] |
| 9 | 7 | 2016 | [35] |
| 10 | 7 | 2018 | [36] |
| 11 | 6 | 2019 | [37] |
| 12 | 6 | 2017 | [38] |
| 13 | 6 | 2019 | [39] |
| 14 | 6 | 2020 | [40] |
| 15 | 6 | 2019 | [41] |

3.2.2. Author Co-Citation Analysis

The distribution of authors with a greater number of citations in that specific field of study was also examined using the author co-citation analysis, which is used to identify the most productive authors in a field. Additionally, the co-citation analysis allows for visualizing similar authors' research areas and subject areas. The author's co-citation analysis for the study on ARS for scab detection in wheat was completed, and the resulting visualization network is shown in Figure 5. The connecting lines between two nodes demonstrate their co-citation relationship, whereas the nodes represent the authors. The number of citations for a given author in the network increases as a node's size increases, making that author more important.

Similarly, the distance between two consecutive nodes or authors is inversely correlated with how frequently each author is cited by the other. The research interests of these authors are more closely correlated with the size of the gap between the nodes. Detailed analysis of the visualization network reveals that the degree of collaboration among most authors is very good, as justified by author co-authorship analysis. The modularity Q and weighted mean silhouette S were 0.6262 and 0.8854, respectively, for the author's co-citation analysis.

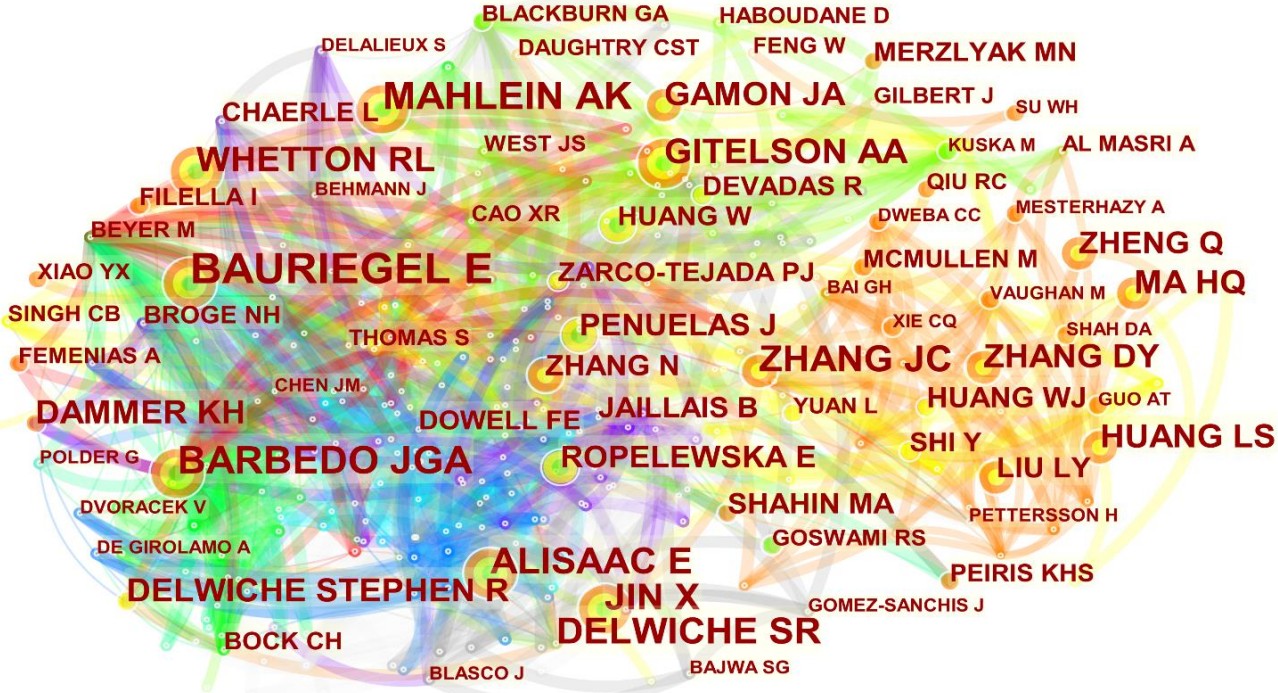

**Figure 5.** The visualization map for the author co-citation network of literature related to agricultural remote sensing for wheat scab detection.

The top fifteen highly co-cited authors are ranked concerning the citation counts of their publications and are listed along with counts of citations, year of citation counts, and the respective authors in Table 10. The statistics reveal that the mentioned authors' work contributed critically to the field of ARS for wheat scab detection, making them highly influential contributors to the upcoming development of agricultural disease detection research. From the results, the authors, including BAURIEGEL, MAHLEIN, and BARBEDO, were the most prolific in the research domain.

**Table 10.** Top 15 most effective co-cited authors of the agricultural remote sensing for wheat scab detection.

| Sr. No. | Count | Year | Cited Authors |
|---|---|---|---|
| 1 | 21 | 2010 | BAURIEGEL E |
| 2 | 16 | 2019 | MAHLEIN AK |
| 3 | 16 | 2017 | BARBEDO JGA |
| 4 | 14 | 2019 | ZHANG JC |
| 5 | 13 | 2006 | DELWICHE SR |
| 6 | 13 | 2019 | ALISAAC E |
| 7 | 13 | 2019 | JIN X |
| 8 | 11 | 2019 | GITELSON AA |
| 9 | 11 | 2019 | WHETTON RL |
| 10 | 10 | 2020 | HUANG LS |
| 11 | 9 | 2013 | DAMMER KH |
| 12 | 9 | 2020 | ZHANG DY |
| 13 | 9 | 2015 | DELWICHE STEPHEN R |
| 14 | 9 | 2020 | MA HQ |
| 15 | 9 | 2009 | GAMON JA |

*3.3. Co-Occurrence Keywords Analysis*

In an article, keywords give details about the subject or broad category to which it specifically belongs. It also represents the primary information in the research papers. Overall, the hotspots and research frontiers can be identified with the aid of keyword co-occurrence analysis. The keywords with the highest citation bursts represent the hotspots frequently cited over time or will be considered in future research. Figure 6 displays the outcomes of the CiteSpace keyword co-occurrence analysis in the form of a visualization network. The nodes represent the keywords, and the size of each node corresponds to the frequency of co-occurrence of each keyword.

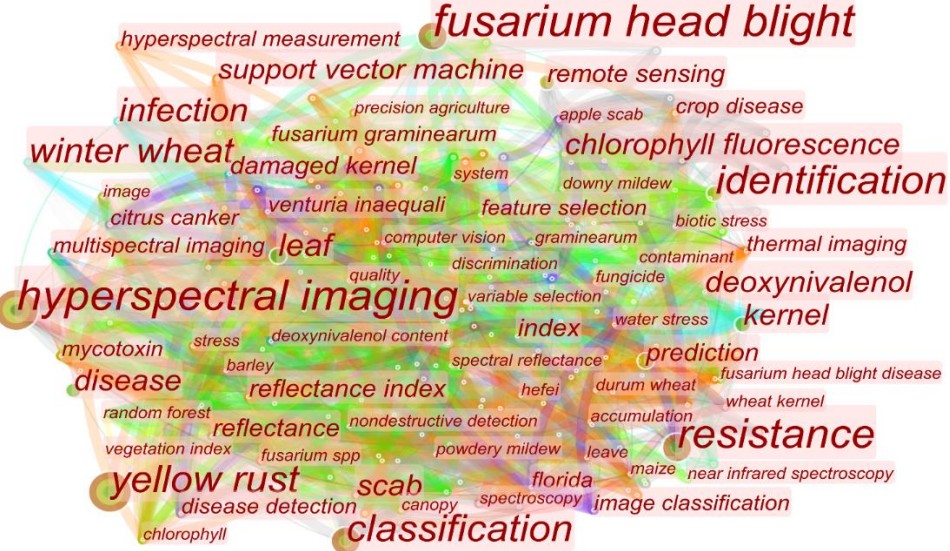

**Figure 6.** The visualization map for the keywords co-occurrence network of agricultural remote sensing for wheat scab detection.

The topmost twenty keywords graded by the number of counts in the field of ARS for wheat scab detection are listed in Table 11. The keywords with the highest co-occurring frequencies (and their counts) were Fusarium head blight (20), hyperspectral imaging (16), resistance (15), and identification (13).

**Table 11.** Top 20 effective keywords of agricultural remote sensing for wheat scab detection ranked by usage frequency.

| Ranking | Counts | Year | Keywords |
|:---:|:---:|:---:|:---:|
| 1 | 20 | 2006 | Fusarium head blight |
| 2 | 16 | 2009 | Hyperspectral imaging |
| 3 | 15 | 2009 | Resistance |
| 4 | 13 | 2009 | Identification |
| 5 | 11 | 2019 | Yellow rust |
| 6 | 11 | 2011 | Classification |
| 7 | 9 | 2006 | Winter wheat |
| 8 | 8 | 2006 | Infection |
| 9 | 7 | 2019 | Leaf |
| 10 | 6 | 2006 | Deoxynivalenol |
| 11 | 6 | 2009 | Chlorophyll fluorescence |
| 12 | 6 | 2010 | Scab |
| 13 | 6 | 2006 | Kernel |
| 14 | 6 | 2016 | Disease |
| 15 | 5 | 2019 | Support vector machine |
| 16 | 4 | 2010 | Reflectance |
| 17 | 4 | 2019 | Reflectance index |
| 18 | 4 | 2015 | Damaged kernel |
| 19 | 4 | 2009 | Index |
| 20 | 4 | 2020 | Prediction |

### 3.4. Hotspots and Research Frontiers

The citation frequency analysis provides a brief overview of the use of the most frequently used keywords in a given time period and can be used to plot these keywords on a time scale for the time period in which these keywords were most frequently used and cited. In the current scenario, citation burst analysis based on co-occurring keywords was performed to determine the hotspots and research frontiers in ARS for wheat scab detection. Hotspots can be recognized as the subfields that are studied most frequently during the development stage of a research area. On the other hand, the keywords that can probably be the most interesting and demanded research fields in the future can be regarded as research frontiers. The conditions, including the number of states, minimum duration, and detection model configuration, used to determine the research limits are shown in Figure 7.

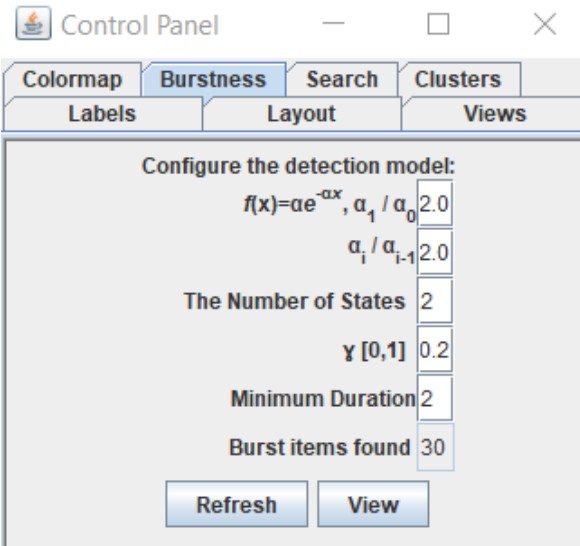

**Figure 7.** Conditions used to evaluate the burst keywords.

In Table 12, from the blue–red-colored timeline analysis, it can be noticed that all 26 keywords with the highest citation bursts were considered by the researchers for small time periods between 2009 and 2022. The studies are being conducted in the research areas related to these keywords. The first nine burst keywords can be regarded as hotspots that were significantly studied before 2022. The last seven burst keywords are the research frontiers that are still in their infancy or need further efforts for their research enhancements.

**Table 12.** Top 26 keywords with the most robust citation bursts in the field of agricultural remote sensing for scab detection.

| Keywords | Strength | Begin | End | 2009–2022 |
|---|---|---|---|---|
| [1] Citrus canker | 1.47 | 2009 | 2013 | |
| [5] Florida | 1.47 | 2009 | 2013 | |
| [1] Biotic stress | 1.07 | 2009 | 2011 | |
| [1] Venturia inaequali | 0.97 | 2009 | 2013 | |
| [1] Scab | 1.31 | 2010 | 2015 | |
| [2] Reflectance | 1.19 | 2010 | 2011 | |
| [1] Apple scab | 1.17 | 2011 | 2013 | |
| [2] Image classification | 1.13 | 2011 | 2018 | |
| [2] Multispectral imaging | 1.08 | 2011 | 2015 | |
| [1] Disease detection | 1.06 | 2011 | 2013 | |
| [1] Deoxynivalenol content | 0.98 | 2015 | 2018 | |
| [1] Damaged kernel | 0.9 | 2015 | 2017 | |
| [1] Graminearum | 0.7 | 2015 | 2019 | |
| [1] Infection | 0.75 | 2016 | 2017 | |
| [1] Identification | 0.82 | 2017 | 2018 | |
| [3] Kernel | 1.49 | 2018 | 2019 | |
| [1] Fusarium head blight disease | 0.95 | 2018 | 2019 | |
| [1] *Fusarium graminearum* | 0.74 | 2018 | 2019 | |
| [1] Yellow rust | 1.93 | 2019 | 2022 | |
| [2] Support vector machine | 1.38 | 2019 | 2020 | |
| [3] Spike | 1.21 | 2019 | 2022 | |
| [4] Feature selection | 1.03 | 2019 | 2020 | |
| [1] Fusarium head blight | 1.41 | 2020 | 2022 | |
| [2] Classification | 1.37 | 2020 | 2022 | |
| [1] Crop disease | 0.94 | 2020 | 2022 | |

[1] The disease; [2] application of remote sensing; [3] scale or organ for disease detection; [4] sensitive feature selection; [5] region.

### 3.5. Description of Cluster Analysis

The research papers published in a given journal describe the cutting edge of the fields covered by that journal. In contrast, the references cited in those papers represent the body of knowledge on which those papers are based. Using CiteSpace, the most frequently employed keywords or references can be clustered through cluster analysis and could be helpful in determining the foundation of basic knowledge in the ARS for wheat scab detection research. The cluster analysis was performed based on co-occurring keywords. The results from cluster analysis are comprehensively discussed in Figure 8. The modularity Q and weighted mean silhouette S were 0.6262 and 0.8854, respectively, for the knowledge cluster analysis.

Furthermore, the details, including cluster ID, cluster size, silhouette, years, and the respective LLR labels, are enlisted in Table 13. A total of 9 knowledge clusters were obtained based on co-occurring keywords' information and ranked in inverse chronological order. The fact that each of the nine major clusters has a silhouette greater than 0.8 indicates that the knowledge has been expertly clustered for the purpose of detecting wheat scab using ARS. The latest (in 2018) cluster ranked (#3) was remote sensing, with 27 articles and a silhouette of 0.827, followed by (#1) random forest, with 31 articles and a silhouette of 0.849 in around 2017.

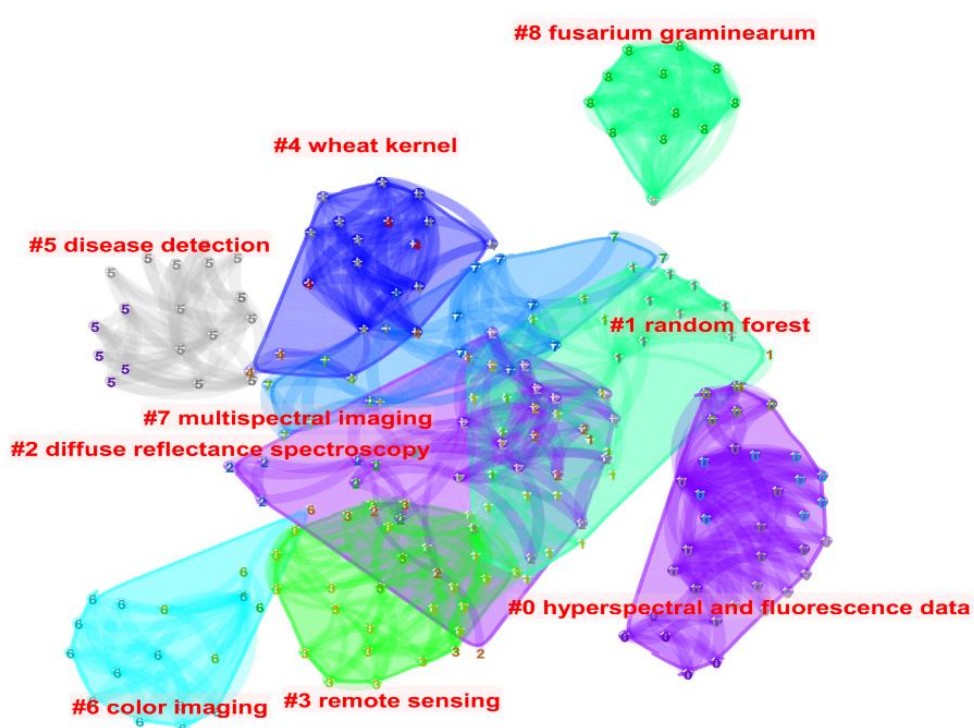

**Figure 8.** Co-occurring keywords-based knowledge clusters of agricultural remote sensing for wheat scab detection based on keywords.

**Table 13.** Details of co-occurring keywords-based knowledge clusters of agricultural remote sensing for wheat scab detection based on keywords.

| Cluster-ID | Size | Silhouette | Year | LLR * Based Keywords |
|---|---|---|---|---|
| 3 | 27 | 0.827 | 2018 | Remote sensing (9.82, 0.005); hyperspectral (6.5, 0.05); precision agriculture (6.5, 0.05); continuous wavelet analysis (6.5, 0.05); feature selection (3.23, 0.1) |
| 1 | 31 | 0.849 | 2017 | Random forest (8.3, 0.005); support vector machine (4.74, 0.05); correlation analysis (4.11, 0.05); fusarium damage (4.11, 0.05); fusion of spectral and image (4.11, 0.05) |
| 8 | 12 | 0.987 | 2017 | *Fusarium graminearum* (7.02, 0.01); fungicide resistance (7.02, 0.01); benzimidazole fungicides (7.02, 0.01); fusarium asiaticum (7.02, 0.01); loop-mediated isothermal amplification-fluorescent loop primer (7.02, 0.01) |
| 6 | 17 | 0.909 | 2016 | Color imaging (6.14, 0.05); potato (6.14, 0.05); fluorescence resonance energy transfer (6.14, 0.05); hybprobes (6.14, 0.05); common scab pathogens (6.14, 0.05) |
| 7 | 15 | 0.833 | 2015 | Multispectral imaging (4.58, 0.05); optical wavelength selection (4.03, 0.05); visualization map (4.03, 0.05); band selection (4.03, 0.05); plant disease (4.03, 0.05) |
| 4 | 19 | 0.886 | 2014 | Wheat kernel (4.03, 0.05); early detection (4.03, 0.05); toxigenic fungi (4.03, 0.05); near-infrared spectroscopy (4.03, 0.05); food commodities (4.03, 0.05) |
| 0 | 35 | 0.966 | 2013 | Hyperspectral and fluorescence data (6.39, 0.05); oculimacula spp (6.39, 0.05); ojip (6.39, 0.05); chlorophyll fluorescence (6.39, 0.05); scab detection (6.39, 0.05) |
| 2 | 30 | 0.826 | 2013 | Diffuse reflectance spectroscopy (3.17, 0.1); hyperspectral image (3.17, 0.1); early disease detection (3.17, 0.1); photosynthesis (3.17, 0.1); flour (3.17, 0.1) |
| 5 | 18 | 0.931 | 2010 | Disease detection (7.55, 0.01); citrus canker (7.55, 0.01); hyperspectral reflectance imaging (5.54, 0.05); lesion size (5.54, 0.05); spectral similarity (5.54, 0.05) |

* LLR abbreviation of log-likelihood ratio used to achieve the optimal results with maximum coverage and uniqueness.

Although disease effects can vary yearly, they are always present and can be a significant challenge even if they only infect certain plant parts. All parts of a plant are susceptible to disease, and multiple diseases can infect the same plant simultaneously; as long as host cultivars and environmental conditions are favorable, they can appear in any field. Scab and three types of rust fungi have historically caused significant crop losses. They remain economically important despite the widespread use of fungicides and host resistance cultivars. However, many pests and diseases are known to reduce grain quality and yield potential [21]. Therefore, scab attracts attention because of its widespread distribution and severe impact on grain quality. However, the bibliometric review and analyses of scab under ARS demand the significant attention of the scientific community. Although the recent focus has been devoted to scab [40,42–45] and other plant diseases, a conclusive methodology at a large scale is still highly needed.

### 3.6. New Trends and Recent Research Status in the Field of INISWS

The following are some of the study's most important findings:

- Regarding INISWS research, the most productive authors at the micro level are Jin X, Alisaac E., Barbedo J.G.A., Ropelewska, Zhang N., Ma H.Q., and others. Researchers who have been cited frequently in INISWS include Bauriegel E., Mahlein A.K., Barbedo J.G.A., Zhang J.C., and others.
- At the meso level, the Chinese Academy of Sciences, Anhui University, and the United States Department of Agriculture are the most active and effective contributors to INISWS research.
- At the macro level, China, the United States, Germany, Italy, Canada, France, and England are the most active and effective contributors to INISWS research. China and the United States have a much higher number of publications than the rest of the countries on the list, and the most likely explanation for this is the more robust funding support policy from both governments. The National Natural Science Foundation of China (NSFC), National Key Research and Development Program of China, National Key R D Program of China, UK Research Innovation, and others have provided the most funding for INISWS research.
- In terms of core journals, the most valuable publications that contributed were: *Remote Sensing*, *Computers and Electronics in Agriculture*, *Frontiers in Plant Science*, and *Biosystems Engineering*.
- The essential knowledge clusters under CiteSpace analysis were hyperspectral and fluorescence data, random forest, diffuse reflectance spectroscopy, and remote sensing.
- The hot research topics were crop disease, identification, feature selection, fusarium head blight, and classification.
- Recent advancements in scab monitoring or detection still need to produce conclusive findings, which are essentially needed.

Conclusively for INISWS researchers, the above results provide important information on new trends and the most recent research status in the field.

### 3.7. Scab Examination in Wheat Using ARS

Scab is a devastating spike disease that has appealed the researchers to devise a remedy for the quality and quantity concerns. Thus, several worldwide scientists worked on this, using different ARS approaches. In this section, we reviewed the summary of conducted investigations for scab detection in wheat kernels (Figure 9A), spikes (Figure 9B), and canopy scale (Figure 9C,D). The below-mentioned studies were conducted using imaging (hyperspectral imaging, fluorescence imaging) and non-imaging (hyperspectral reflectance) instruments. This section highlights the conclusive findings about Scab disease in ARS considering the wavelength range, sensors, sensitive band selection algorithms, classification algorithms, and sensitive bands for different studies conducted in different countries.

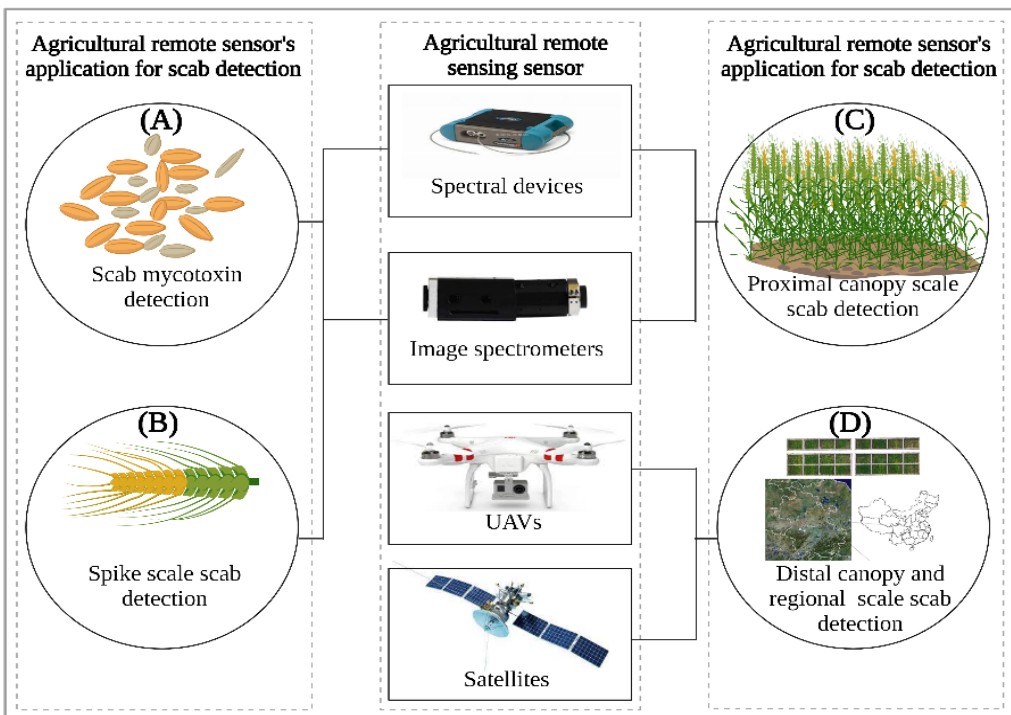

**Figure 9.** Agricultural remote sensors and their application for scab detection (**A**) to quantify the mycotoxin quantity in wheat seeds, (**B**) detect the wheat scab through examination of wheat spike, (**C**) detect and monitor the scab at proximal canopy scale, and (**D**) monitor the scab at canopy and regional scale.

### 3.7.1. ARS for Scab Detection in Wheat Kernels

Table 14 enlists the studies conducted for INISWS detection in wheat kernels. Numerous studies have been conducted for scab detection because grain is the most useful part of human and livestock consumption, directly affecting food and feed quality. Considering INISWS, different algorithmic and sensors are employed for the non-destructive monitoring of plants. Hyperspectral image (HSI) analysis with ratio analysis (RA) selected sensitive bands 568 and 715 nm and manifested high classification accuracy with linear discriminant analysis (LDA) [46]. In another study, texture features classified the healthy and infected grains with PCA and support vector machine (SVM) [47]. In recent studies [48–51], HSI and multispectral images were employed to discriminate between the scab-infected and healthy grains through sensitive feature selection and machine learning classifiers. The application of machine learning classifiers for discrimination among healthy and treated plants has become the top-notch approach for the ARS community [35].

**Table 14.** Summary of the investigations for scab detection in wheat kernels using ARS technology.

| Wavelength Range (nm) | Spectrometer (Sensor) | Sensitive Band Selection Approach | Discriminant and Estimation Algorithms | Sensitive Bands (nm) | Location | References |
|---|---|---|---|---|---|---|
| 425–860 | HSI | RA | LDA | 568, 715 | USA | [46] |
| 900–1700 | HSI | PCA | SVM | Texture features | Canada | [47] |
| 1000–1600 | NIR-HSI | PCA | LDA, QDA, MD | 1284, 1316, 1347 | Canada | [52] |
| 1000–1700 | NIR-HSI | LMM | LDA | 1002, 1127, 1199, 1315, 1474 | USA | [53] |
| 400–1000 | NIR-HSI | PCA | LDA, PCA | 484, 567, 684, 818, 900, 950 | Canada | [54] |
| 400–1700 | NIR-HSI | RA | LDA | 502, 678, 1198, 1496 | USA | [55] |

**Table 14.** *Cont.*

| Wavelength Range (nm) | Spectrometer (Sensor) | Sensitive Band Selection Approach | Discriminant and Estimation Algorithms | Sensitive Bands (nm) | Location | References |
|---|---|---|---|---|---|---|
| 400–1700 | HSI | PCA | LDA, QDA, MD | 870 | Canada | [56] |
| 400–1000 | HSI | PLSR | PLS-DA | 450, 494, 578, 639, 678, 717, 819, 853, 883, 903, 917, 942, 950 | Canada | [57] |
| 1000–1700 | HSI | PCA, PLS-DA, iPLS-DA | PLS-DA | 1209–1230, 1489–1510, 1601–1622 | Italy | [58] |
| 360–950 | HSI | PCA | URA | 875, 950 | France | [59] |
| 528–1785 | HSI | PCA | LDA | 672, 1361, 1411, 1509, 1657 | Brazil | [30] |
| 1000–1600 | NIR-HSI | PCA | MD, LDA, QDA | 1280, 1300, 1350 | Canada | [60] |
| 820–1666 | NIR-HSI | GA | ICA | | Canada | [61] |
| 405–970 | MS | PCA, | Knn | 590–890 | Denmark | [62] |
| 528–1785 | HSI | | PC | 623, 672, 1361, 1411, 1509, 1657 | Brazil | [38] |
| 866.4–1701.0 | HSI | PCA | PLS-DA, SVM, Knn | 1105.3, 1199.2, 1305.3, 1321.7, 1439.3, 1458.7, 1478.1 | China | [63] |
| 400–2500 | HSI | COR | | 538–572, 828–1000, 1350–2500 | Germany | [64] |
| 400–1000 | HSI | PCA, SPA, RF | SVM, RF, NB | 513, 754, 836, 849, 860, 880 | China | [65] |
| 900–1700 | HSI | PCA | PLS and LDA | 955, 1278, 1403, 1455, 1528, 1671, 1714 | Spain | [66] |
| 400–2500 | HSI | GA | SVM, SAE | 570–710, 1050–1089, 1128–1313, 1666–1744, 1005, 1403, 1843, 1879, 1912, 1980 | China | [67] |
| 350–2500 | HR | SPA | PLS-DA, SVM | 1878, 1887 | China | [68] |
| 374–1030 | HSI | R-Frog | Knn, CNNs | 940, 678, 728, 798, 1009 | China | [69] |
| 900–1700 | HSI-NIR | PLS, LDA | PLS, LDA | 1220, 1380 | Spain | [70] |
| 960–1700 | HSI | | Knn | Whole spectra | Canada | [71] |
| 900–1700 | HSI-NIR | PCA | LDA, NB, PLSR | 1325, 1396, 1406, 1421 | Canada | [72] |
| 940–1600 | HSI | PCA, | LDA | 986, 1000, 1111, 1197, 1394, 1200, 1260, 1460 | USA | [73] |
| 900–1700 | HSI-NIR | PCA | PLS, SVM, local PLS | 970, 1200, 1365, 1430, 1623 | China | [50] |
| 374–1030 | HSI | Relief F, R-Frog, shuffled frog | KNN, SVM, CNN, LeNet, VGG-16 | 732, 876, 941, 988 | China | [51] |
| 866–1701 | HSI-NIR | DCGAN | CNN, SVM, DT | 1150–1300, 1400–1650 | China | [48] |
| 405–970 | MS | PCA, GA | SVM, PLS, BPNN | 910, 910–970 | China | [49] |

HSI: hyperspectral images; RA: ratio analysis; LDA: linear discriminant analysis; USA: united states of America; PCA: principal component analysis; SVM: support vector machine; NIR-HSI: near-infrared hyperspectral images; QDA: quadratic discriminant analysis; MD: Mahalanobis discriminant; LMM: local minima or maxima; PLSR: partial least square regression; PLS-DA: partial least squares discriminant analysis; iPLS-DA: interval partial least squares discriminant analysis; URA: univariate regression analysis; GA: genetic algorithm; ICA: independent component analysis; MS: multispectral; Knn: k-nearest neighbor; PC: probabilities calculation algorithm; SPA: successive projection algorithm; RF: random forest; COR: correlation; NB: Naive Bayes; SAE: sparse autoencoder; R-Frog: random frog; CNN: convolution neural networks; HR: hyperspectral reflectance; DCGAN: deep convolutional generative adversarial network; DT: decision tree; PLS: partial least square; BPNN: back propagation neural network.

### 3.7.2. ARS for Scab Detection in Wheat Spikes

Table 15 shows the application of imaging and non-imaging spectrometers for scab detection at spike scale in different studies and highlights the sensitive bands of different sensors and best algorithmic approaches for scab detection at spike scale. Principal component analysis (PCA) and spectral angle mapper (SAM) for the feature selection and classification were used by [74], employing HSI of scab disease in wheat crops. In addition, they also compared HSI with chlorophyll fluorescence using SAM for the classification-wavelength range 400–1000 nm [75]. In other findings, the PLS-DA model was used with a Euclidean distance matrix cladogram to classify the diseased and healthy spikes at different severity levels. The VIS-NIR spectral analysis could facilitate the detection of scab [76]. SVM for classification with non-metric multidimensional scaling (NMDS) to analyze different vegetation indices is a good approach for scab disease detection [23]. In HSI, a hybrid two-dimensional convolutional bidirectional gated recurrent unit neural network (2D-CNN-BidGRU) has an accuracy of 0.75 and 0.743 for classifying the diseased pixels compared to healthy ones [29]. Moreover, [39] compared infrared thermography (IRT), chlorophyll fluorescence imaging (CFI), and HSI techniques to relate the temperature, stress and spectral response of diseased spikes against healthy spikes with an accuracy of 78, 56, and 78%, respectively. Combining the IRT-HSI or CFI-HSI parameters improved the accuracy to 89% three days after inoculation. All the scientists acquired good results in scab detection and classification, but there was a lack of early detection approaches. Nevertheless, a reliable method of scab detection has yet to be developed.

**Table 15.** Summary of the investigations for scab detection in wheat spikes using ARS technology.

| Wavelength Range (nm) | Spectrometer | Sensitive Band Selection Approach | Discriminant and Estimation Algorithms | Sensitive Bands (nm) | Location | References |
|---|---|---|---|---|---|---|
| 620 | FluorCam 700 MF | | URA | | Czech Republic | [77] |
| 400–1000 | HSI | | PCA, SAM | 560–560, 665–675 | Germany | [74] |
| 400–1000 | HSI | PCA | DCNN, DCRNN, DRNN | 670, 665–675 | China | [29] |
| 400–2500 | HSI | MDS | SVM | 430–525, 560–710, 1115–2500 | Germany | [23] |
| 400–1000 | HR | FLDA | SVM, LDA | First order derivatives (490–530, 510–530) | China | [34] |
| 400–2400 | IRT, CFI, HSI | COR | SVM | 500, 675, 760, 1440, 1880, 2000 | Germany | [39] |
| 400–1000 | HSI | ISI | SAM | 539, 417, 468 | China | [32] |
| | RGB | | DCNN | KMC, Otsu's method | China | [78] |
| 400–1000 | HR | | URA | 450–488, 500–540 | China | [43] |
| 400–900 | HSI | COR, Relief F, RF, SFS-FS, SVM-RFE, LASSO-LR | QDA | 540, 591, 696, 766, 868 | USA | [79] |
| | RGB | | PCNN, KMC | | China | [80] |
| 374–1050 | HSI | RF | URA | 560, 565, 570, 661, 663, 678 | China | [81] |
| 400–1000 | HR | CWT, COR | SVM, GA-SVM | MSR, SIPI, NDVI | China | [82] |

**Table 15.** *Cont.*

| Wavelength Range (nm) | Spectrometer | Sensitive Band Selection Approach | Discriminant and Estimation Algorithms | Sensitive Bands (nm) | Location | References |
|---|---|---|---|---|---|---|
| 400–1000 | HSI | SPA, COR | PSO-SVM | 442, 491, 552, 675, 685, 693, 698, 706, 757, 767, 924, 935 | China | [40] |
| 400–2500 | HR | CWA | FLDA | 471, 696, 841, 963, 1069, 2272 | China | [9] |
| 400–1000 | HSI | PCA, GB, DT | DCNN, RF, PLSR, SVR | 480, 560, 660 | China | [83] |
| | RGB | | Mask-RCNN | IS | China | [84] |
| | RGB | | Mask-RCNN | IS | China | [85] |
| 400–1000 | HR | CWT | PSO-SVM, RF, BPNN | 474, 495, 528, 582, 615, 691, 738 | China | [86] |
| | RGB | | DCNN | IS | United States | [87] |
| | RGB | Relief-F | RF | | China | [88] |
| 400–1000 | HR | COR | SVM | 561, 562, 563, 581, 582, 585, 590, 597, 598, 599 | China | [44] |
| 400–1000 | HR and CFI | Boruta | KNN, SVM, RF | Chlorophyll indices | China | [89] |

SAM: spectral angle mapper; DCNN: deep convolutional neural network; DCRNN: deep convolutional recurrent neural network; DRNN: deep recurrent neural network; MDS: multidimensional scaling; FLDA: Fisher's linear discriminant analysis; IRT: infrared thermography; CFI: chlorophyll fluorescence imaging; ISI: instability index; SFS-FS: sequential feature selection—forward selection; SVM-RFE: support vector machine recursive feature elimination; LASSO-LR: LASSO logistic regression; PCNN: pulse-coupled neural network, KMC: K-means clustering; GA-SVM: genetic algorithm-SVM; CWT: continuous wavelet transform; MSR: modified simple ratio; SIPI: structure intensive pigment index; NDVI: normalized difference vegetation index; PSO-SVM: particle swarm optimization-SVM; GB: gradient boost; DT: decision tree; SVR: support vector regression; Mark-RCCN: Mask region convolutional neural network; IS: image segmentation.

### 3.7.3. ARS for Scab Detection in the Wheat Canopy

Table 16 demonstrates the ARS community's focus on employing imaging and non-imaging spectrometers to detect wheat scab at the canopy scale, which is unfortunately very limited in a number of studies. Given the devastating threat to food quality and quantity posed by scab, the proxy approach to scab monitoring, classification, detection, and quantification has yet to be studied and finalized. A recent study [45] examined the canopy scale infection of the scab through the development of scab-specific indices using continuous wavelet transform (CWT) and robustified their relevance using different machine learning algorithms (RF, Knn, SVM, NN, Xgboost). Another study used the whole spectral (400–2400 nm) analysis to quantify the scab disease at the canopy scale using SVM algorithmic approach [90]. Likewise, partial least squares regression (PLSR) with PCA at canopy level disease detection of scab and yellow rust diseases by HSI showed satisfactory results [36].

**Table 16.** Summary of the investigations for scab detection in wheat canopy using ARS technology.

| Wavelength Range (nm) | Sensor | Sensitive Band Selection Approach | Discriminant and/or Estimation Algorithms | Sensitive Bands (nm) | Location | References |
|---|---|---|---|---|---|---|
| 400–700 | | MANOVA, PCA | PLSDA | SA | Italy | [76] |
| 400–730 | | PCA | PLSR | 500–650, 650, 700 | United Kingdom | [36] |

**Table 16.** *Cont.*

| Wavelength Range (nm) | Sensor | Sensitive Band Selection Approach | Discriminant and/or Estimation Algorithms | Sensitive Bands (nm) | Location | References |
|---|---|---|---|---|---|---|
| 400–730 | RGB | | PLSR | On field | United Kingdom | [33] |
| | RGB | | KMS | | | [91] |
| | MS | | URA | NDVI, RVI, DVI | China | [92] |
| (400–100) | MS, HR | | URA, OLS | 665, 783, 842 | China | [93] |
| | MODIS | | DTM, RVM | | China | [94] |
| 450–950 | HSI | backward feature selection, | URA, PLSR, FLDA, LR, RF, SVM, BPNN | 650, 670, 690, 730, 770 | China | [95] |
| 400–2400 | HR | | SVM | Spectral analysis | Czech Republic | [90] |
| 450–950 | HSI | | Logistic model | 550, 670, 702, 740 | China | [96] |
| 400–2400 | HR | CWT | RF, Knn, SVM, NN, Xgboost | 401, 460, 789, 840 | China | [45] |
| 450–950 | | RF | RF, BPNN, SVM | 518, 666, 706, 742, 846 | China | [97] |

MANOVA: multivariate analysis of variance; SA: spectral analysis; NN: neural net; Xgboost: extreme GB; RVI: ratio vegetation index; DVI: difference vegetation index; OLS: ordinary least square; DTM: decision tree model; RVM: relevance vector machine.

### 3.7.4. Quantitative Models for Scab Disease

Numerous studies have used univariate and multivariate quantitative models for disease estimation in ARS for diverse plant diseases [98]. Different studies estimated the scab disease at spike [89] and canopy scale [45] using KNN predictive model. The studies also used SVM and RF models, but KNN outperformed the competing models. Moreover, the studies have also concluded that the comparative performance of disease-specific or newly developed indices for model fitting is highly improved against conventional indices [45,89] that support different other studies [99]; based on these results, a precise FHB monitoring program can be developed. Moreover, the better disease estimation performance of models through disease-specific indices or bands has also been proved by previous findings [42,99]. Another study also estimated the canopy scale scab disease considering the optimal window size of texture features [96]. Hence, a separate detailed study can be conducted using raw data and the application of different predictive models for deep exploration of the features.

### 4. Limitations and Future Prospective

Although several sensors have been used for scab detection under ARS, a conclusive methodology or technical instrumental setup pertinent to scab-specific detection or monitoring is still lacking. Tables 14–16 show the suitability as well as the chaotic behavior of different algorithms because selected or extracted scab-specific features (wavelengths) vary under different machine learning algorithms. However, the spectral regions (green, blue, red, NIR) are the same among different studies. Hence, the core future perspective is to focus on specific features that can be generalized for real-time detection, monitoring, and quantifying scab disease from spike to canopy scale. Supportively, the burst keywords' analysis (Table 12—last seven burst keywords) highlights the research frontiers that are in need of further research enhancements.

Regarding reviewing the different scab studies, the future suggestion could be that further research and data analysis is required to extract quantitative information on disease levels. Integration of hyperspectral sensor-based information, such as sensitive bands and vegetation indices, into pesticide application maps is very likely. Even more technological advancement is required for online systems. It is needed to be simplified with automated calibration and processing to compensate for different plant parameters, suitable for specialists and non-specialists, and for practical applications of sensor systems and algorithms to analyze hyperspectral data.

## 5. Conclusions

This study conducted a scientometric analysis of the scientific literature on imaging and non-imaging spectroscopy for wheat scab (INISWS). INISWS research published between 2005 and 2022 was extracted from the Web of Science (WOS) using co-citation, co-authorship, and co-occurrence analysis of keywords. The new knowledge structures, developments, authors or institutional collaborations, hot topics, and research frontiers in INISWS-related research were all taken into consideration. However, despite the significant findings from the visualization analysis of INISWS-related articles, the current study has a number of shortcomings. Only English-language publications are included in the WOS core collection databases, so there is a small amount of residue in terms of citations.

In comprehensive conclusion, while remote sensing as a technique provides the potential for high-accuracy diagnosis, monitoring, and management of scab diseases in wheat, there exists a need for further research with a focus on identifying scab-specific features applicable across sensors and algorithms. This would enable more accurate and reliable detection, monitoring, and quantification of scab, thereby improving crop management practices, mitigating economic losses, and ensuring food security. Integration of hyperspectral sensor-based information into pesticide application maps, accompanied by technological advancement for simplified and automated processing and calibration, is envisaged as an essential step to enable practical applications of sensor systems and algorithms to analyze hyperspectral data in wheat crops. However, to develop comprehensive decision support systems, research needs to focus on extracting quantitative information on disease levels using hyperspectral data. To achieve reliable and accurate data results, researchers must use appropriate sensors, concentrate on green, blue, red, NIR spectral regions, and develop machine learning algorithms tailored explicitly for scab disease detection. With these developments, remote sensing and analysis of scab disease will greatly help improve agricultural productivity worldwide.

**Author Contributions:** Conceptualization, I.A., M.C., G.M. and S.H.; methodology, G.M., M.C. and I.H.K.; software, G.M., J.L., C.C. and B.H.; validation, S.H., I.H.K. and Y.L.; formal analysis, G.M. and I.H.K.; investigation, S.H., J.L., C.C., I.A. and B.H.; supervision, Y.L.; project administration, Y.L.; funding acquisition, Y.L. and S.H. All authors have read and agreed to the published version of the manuscript.

**Funding:** This research is funded by the Jiangsu Province Postdoctoral Excellence Program (2022ZB155), Science and Technology Major Project of Inner Mongolia (No. ZDZX2018054), China, and by the National Natural Science Funds of China (31370474).

**Data Availability Statement:** Not applicable.

**Acknowledgments:** All authors contributed equally in this manuscript. All authors read and approved the submitted version.

**Conflicts of Interest:** The authors declare that the research was conducted in the absence of any identification of Fusarium Head Blight in Winter Wheat Ears Using Continuous Wavelet Analysis.

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
