# Peer review of "Global Trends and Future Directions in Agricultural Remote Sensing for Wheat Scab Detection: Insights from a Bibliometric Analysis"

_remotesensing, doi:10.3390/rs15133431_

Round 1
Reviewer 1 Report
very good
very good
Author Response
Dear esteemed reviewer,
I hope this message finds you well. I would like to express my deepest gratitude for taking the time to review my work. Your feedback and suggestions have been extremely valuable, and I appreciate your willingness to share your expertise with me.
Thanks for appreciation, I carefully considered your comments and have revised again carefully. Your insights have helped me to improve the quality of my work, and I feel confident that it is now much stronger as a result of your input.
Once again, thank you for your time and attention. I look forward to any future opportunities to work with you and benefit from your thoughtful feedback.
Best regards,
Dr. Ghulam Mustafa
Comments
Very Good
Response:
Thanks for appreciation and motivation

Reviewer 2 Report
The research is too insufficient to summary the position, mechanism, and prevention and control measures of Wheat Scab. The research also lacks a systematic summary of future research directions.
Author Response
Dear Reviewer,
I am writing to express my appreciation for your detailed and insightful review of my work. I found your comments and suggestions to be incredibly helpful in improving the overall quality of my project. Your expertise and attention to detail have been invaluable, and I am grateful for the time and effort you put into reviewing my work.
I sincerely appreciate your willingness to share your feedback with me, and I am confident that the changes made now and suggestion for future will greatly enhance the final outcome. Thank you once again for your assistance, and I look forward to continuing to work with you in the future.
Best regards,
Dr. Ghulam Mustafa
Comments
The research is too insufficient to summary the position, mechanism, and prevention and control measures of Wheat Scab. The research also lacks a systematic summary of future research directions.
Response:
Respected reviewer, since this study focused on Bibliometric review of previous studies, we did not ensure much work on the position, mechanism etc. However, this is a great suggestion we will focus in detail for future studies.
In addition, we have added further suggestions for future research directions under agricultural remote sensing (ARS).
- Limitations and future prospective
Although several sensors have been used for scab detection under ARS, a conclusive methodology or technical instrumental setup pertinent to scab specific detection or monitoring is still lacking. Tables 14 - 16 show the suitability as well as chaotic behavior of different algorithms because selected or extracted scab specific features (wavelengths) vary under different machine learning algorithms. However, the spectral regions (Green, Blue, Red, NIR) are same among different studies. Hence, the core future perspective is to focus on specific features that can be generalized for real-time detection, monitoring, and quantifying scab disease from spike to canopy scale. Supportively, the burst keywords’ analysis (Table 12 – last 7 burst keywords) highlights the research frontiers that are in need of further research enhancements.
Regarding reviewing the different scab studies, the future suggestion could be that the further research and data analysis are required to extract quantitative information on disease levels. Integration of hyperspectral sensor-based information, such as sensitive bands and vegetation indices into pesticide application maps, is very likely. Even more technological advancement is required for online systems. It is needed to be simplified with automated calibration and processing to compensate for different plant parameters, suitable for specialists and non-specialists, and for practical applications of sensor systems and algorithms to analyze hyperspectral data.

Reviewer 3 Report
Do you preprocess the spectral data and characteristic wavelengths whether extracted using any spectral preprocessing methods . Also enlighten on spectral quantitative prediction models for the infection degree of Scab in the in situ wheat canopy visible areas. This can be established using the PLSR method, based on the original spectral data, preprocessed spectral data, original spectral characteristic wavelengths extracted data, and preprocessed spectral characteristic wavelengths extracted data. Look in to the possibilities for further exploration based on the above comments.
Author Response
Dear Reviewer,
I would like to extend my deepest thanks for taking the time to review my work and provide such thoughtful feedback. Your insights and suggestions have helped me to refine and strengthen my project, and I am grateful for the opportunity to learn from your expertise.
I am especially impressed by the level of detail in your comments, which demonstrate a deep understanding of the subject matter and a commitment to quality. Your review has been invaluable, and I am confident that it will help me to create a project of the highest standard.
Once again, thank you very much for your valuable feedback. It is truly appreciated, and I look forward to the opportunity to work with you again in the future.
Sincerely,
Dr. Ghulam Mustafa
Comments
Do you preprocess the spectral data and characteristic wavelengths whether extracted using any spectral preprocessing methods.
Also enlighten on spectral quantitative prediction models for the infection degree of Scab in the in-situ wheat canopy visible areas.
This can be established using the PLSR method, based on the original spectral data, preprocessed spectral data, original spectral characteristic wavelengths extracted data, and preprocessed spectral characteristic wavelengths extracted data.
Look in to the possibilities for further exploration based on the above comments.
Response:
Thanks for further suggestion,
The current study didn’t use the raw data for wavelength selection or development of quantitative models. We reviewed the previously conducted studies and highlighted the sensitive wavelengths to wheat scab disease. Supportively we have enlightened on spectral quantitative prediction models for the infection degree of Scab in the in-situ wheat canopy visible areas.
3.6.4 Quantitative models for Scab disease
Numerous studies have used the univariate and multivariate quantitative models for disease estimation in ARS for diverse plant diseases [95]. Different studies estimated the scab disease at spike [84] and canopy scale [34] using KNN predictive model. The studies also used SVM and RF models but KNN outperformed the competing models. Moreover the studies have also concluded that the comparative performance of disease specific or newly developed indices for model fitting is highly improved against conventional indices [34,84] that support different other studies [96], based on these results a precise FHB monitoring program can be developed. Moreover, the better disease estimation performance of models through disease specific indices or bands has also been proved by previous finding [30,96]. Another study also estimated the canopy scale scab disease considering the optimal window size of texture features [93]. Hence, a separate detailed study can be conducted using raw data and application of different predictive models for deep exploration of the features.

Reviewer 4 Report
The review entitled “Global Trends and Future Directions in Imaging and Non-imaging Spectroscopy for Wheat Scab Detection: Insights from a Bibliometric Analysis” is a very interesting one, including 93 references, many tables and figures and is a very clearly and easy to read manuscript.
Good luck!
Author Response
Dear Reviewer,
I wanted to take a moment to express my sincere gratitude for your kind and constructive feedback on my work. Your comments were extremely helpful in identifying areas for improvement, and I appreciate the time and effort you put into reviewing my project.
Your professional insights, appreciation and attention to detail have been instrumental in ensuring that my work meets the highest standards. Without your valuable input, I would not have been able to achieve the level of clarity and effectiveness that I now feel confident in.
Thank you once again for your assistance, and I look forward to future opportunities to collaborate on meaningful projects.
Warmest regards,
Dr. Ghulam Mustafa
Comments
The review entitled “Global Trends and Future Directions in Imaging and Non-imaging Spectroscopy for Wheat Scab Detection: Insights from a Bibliometric Analysis” is a very interesting one, including 93 references, many tables and figures and is a very clearly and easy to read manuscript.
Good luck!
Response:
Thanks for appreciation and motivation.
